

# ULK2 suppresses ovarian cancer cell migration and invasion by elevating IGFBP3

Xiaoxi Chen[1,2,*], Changxiang Shao[1,*], Jing Liu[1], Huizhen Sun[3], Bingyi Yao[1], Chengbin Ma[1], Han Xu[4] and Weipei Zhu[2]

[1] Changning Maternity and Infant Health Hospital, East China Normal University, Shanghai, China
[2] The Second Affiliated Hospital of Soochow University, Soochow University, Soochow, Jiangsu, China
[3] Department of Obstetrics and Gynecology, Xinhua Hospital Affiliated to Shanghai Jiaotong University School of Medicine, Shanghai, China
[4] Department of General Surgery, Jing'an District Center Hospital of Shanghai, Shanghai, China
* These authors contributed equally to this work.

## ABSTRACT

**Background:** Ovarian cancer is an aggressive malignancy with high mortality known for its considerable metastatic potential. This study aimed to explore the expression and functional role of Unc-51 like autophagy activating kinase 2 (ULK2) in the progression of ovarian cancer.
**Methods:** ULK2 expression patterns in ovarian cancer tissues as well as benign tumor control samples obtained from our institution were evaluated using immunohistochemistry. Cell counting kit 8 and Transwell assays were applied to assess the effects of ULK2 overexpression on cell proliferation, migration and invasion, respectively. RNA sequencing was performed to explore potential mechanisms of action of ULK2 beyond its classical autophagy modulation.
**Results:** Our experiments showed significant downregulation of ULK2 in ovarian cancer tissues. Importantly, low expression of ULK2 was markedly correlated with decreased overall survival. *In vitro* functional studies further demonstrated that overexpression of ULK2 significantly suppressed tumor cell proliferation, migration, and invasion. RNA sequencing analysis revealed a potential regulatory role of ULK2 in the insulin signaling pathway through upregulation of insulin-like growth factor binding protein-3 (IGFBP3) in ovarian cancer cells.
**Conclusions:** In summary, the collective data indicated that ULK2 acted as a tumor suppressor in ovarian cancer by upregulating the expression of IGFBP3. Our study underscores the potential utility of ULK2 as a valuable prognostic marker for ovarian cancer.

## INTRODUCTION

Ovarian cancer is the primary contributor to mortality among gynecological malignances worldwide (*Chen et al., 2021*), largely due to extensive dissemination to serosal surfaces and concurrent peritoneal metastasis (*Smolle, Taucher & Haybaeck, 2014*; *Nieddu et al.,*

Corresponding authors
Han Xu, 17301793781@163.com
Weipei Zhu, zwp3333@suda.edu.cn

*2023*; *Zhang et al., 2022*). However, the precise mechanisms underlying the metastatic progression of ovarian cancer remain poorly understood, highlighting a critical need to comprehend the molecular pathways implicated in disease development.

Unc-51 like autophagy activating kinase 2 (ULK2), a key serine/threonine protein kinase, is pivotal in the initiation of autophagy in various cancers, including gastric cancer (*Motoo et al., 2022*), lung adenocarcinoma (*Tsang et al., 2020*) and prostate cancer (*Hu et al., 2020*). Furthermore, a mounting number of studies suggest that ULK2 is implicated in cancer progression and therapy by influencing diverse cellular processes independent of its involvement in autophagy, rendering it a putative tumor suppressor gene. For instance, the ULK1/2-paxillin mechanotransduction pathway suppresses the migration of breast cancer cells independently of its role in autophagy (*Liang et al., 2023*). However, the function of ULK2 in ovarian cancer remains poorly understood at present.

The insulin signaling pathway plays a crucial modulatory role in tumorigenesis and tumor advancement (*Solarek et al., 2019*). Additionally, insulin signaling is associated with regulation of cancer-associated fibroblasts (CAFs) in the tumor microenvironment to further promote tumor growth and progression (*Zhai et al., 2023*). Insulin-like growth factor binding protein-3 (IGFBP3) serves as a key component in the insulin signaling pathway. Previous studies have demonstrated that IGFBP3 negatively modulate the insulin secretion and insulin signaling pathway (*D'Addio et al., 2022*), exerting inhibitory effects on tumor cell proliferation and progression (*Zhong et al., 2024*; *Kuhn et al., 2023*).

The study aimed to explore the expression and functional significance of ULK2 in the progression of ovarian cancer as well as mechanisms other than autophagy regulation. Initially, analysis of patient samples revealed significant downregulation of ULK2 in ovarian cancer tissues relative to benign ovarian tumor samples. In subsequent *in vitro* experiments, ULK2 negatively regulated proliferation, motility, and invasion of ovarian cancer cells. Finally, mechanistic studies disclosed that ULK2 upregulated insulin-like growth factor binding protein-3 (IGFBP3), in turn, leading to suppression of the insulin signaling pathway. Therefore, the collective results suggest that ULK2 participates in tumor inhibition and could potentially be utilized as a novel prognostic biomarker for ovarian cancer.

## METHODS

### Patient samples and immunochemistry

Before conducting the experiment, we obtained ethical approval from the Changning District Maternal and Child Health Hospital firstly (No. CNFBLLKT-2023-013). WRITTEN consent was obtained from all participants prior to their participation in the study. Immunohistochemistry (IHC) was utilized to assess ULK2 expression level in 81 benign ovarian tumor and 80 epithelial ovarian cancer tissue microarray (TMA) samples. The essential steps for IHC were summarized as follows. TMA were deparaffinized to remove wax followed by the application of a Hydrogen Peroxide Block (P0100; Beyotime, Shanghai, China) to minimize non-specific background staining caused by endogenous peroxidase. Subsequently, ULK2 primary antibody (PA5-22173; Thermo Fisher, Waltham, MA, USA) was diluted to a ratio of 1:200 and incubated at 37 °C for 1.5 h. After PBS

washing, samples were further incubated with HRP-conjugated secondary antibody at ambient room temperature for 30 min. The microarray was subsequently stained using a DAB horseradish peroxidase color development kit (P0203; Beyotime, Shanghai, China). Finally, stained microarray samples were observed under a microscope and subjected to scanning analysis. H-score was calculated using the formula: H-SCORE = $\sum(pi \times i)$ = (percentage of weak intensity $\times$ 1) + (percentage of moderate intensity $\times$ 2) + (percentage of strong intensity $\times$ 3) (*Maclean et al., 2020*).

## Lentiviral transfection and cell culture

Overexpression of the ULK2 gene was achieved using a lentiviral vector obtained from Genechem Co. (Shanghai, China). The coding sequence of the ULK2 gene was integrated into the lentiviral vector. Cells transfected with scramble were utilized as control. The efficiency of the lentivirus infection was validated by western blot.

Ovarian cancer cell lines (OVCA433 and HEY A8) obtained from American Type Culture Collection (Manassas, VA, USA) were cultured in Dulbecco's Modified Eagle Medium (DMEM) containing 10% Fetal Bovine Serum (FBS) at 37 °C in a controlled environment with 5% $CO_2$.

## Western blot

Protein extraction was performed from the precipitate of ovarian cancer cells and protein concentration was determined with a BCA assay kit (P0010; Beyotime, Shanghai, China). A total of 30 μg protein was loaded into a 10% sodium dodecyl sulfate polyacrylamide gel electrophoresis (SDS-PAGE) gel, followed by transfer to polyvinylidene fluoride (PVDF) membrane and blocking procedures. The PVDF membrane was then incubated with primary antibody overnight at 4 °C. The ULK2 antibody was obtained from Abcam (ab97695, Cambridge, UK). IGFBP3 antibody was purchased from cell signaling technology (25864, MA, USA). β-actin from Proteintech (66009-1-Ig; Wuhan, Hubei, China) was utilized as the internal control. Subsequently, PVDF membrane was treated with the respective secondary antibody at room temperature (RT) for 1 h after 3 times washing with Tris-Buffered Saline with Tween 20 (TBST). Secondary antibodies against rabbit or mouse were acquired from Proteintech (SA00001-1 or SA00001-2). Finally, a chemiluminescence detection kit (P2200; NCM Biotech, Soochow, China) was used to visualize specific proteins.

## Cell counting kit 8 (CCK-8) assay

The cell counting kit 8 (CCK-8) assay was used to evaluate ovarian cancer cell proliferation. Ovarian cancer cells were seeded at a concentration of 1,000 cells per well in 96-well plate. Subsequently, each well was treated with 10 μL CCK-8 reagent with 1 h incubation at 37 °C. Absorbance was monitored at 24, 48, and 72 h with a microplate reader at 450 nm.

## Transwell assay

Transwell migration and invasion analysis was conducted to assess the migratory and invasive abilities of ovarian cancer cells. Ovarian cancer cell lines were maintained in

DMEM with 10% FBS. Once 80% coverage was reached, cells were detached into individual cells using 0.25% trypsin-EDTA solution, and cell counts were determined. Transwell plates with inserts containing porous membranes were prepared in a 24-well plate. Cell solution was generated with DMEM without FBS at a concentration of $1 \times 10^5$ cells/mL. Subsequently, 200 μL cell solution was introduced into each Transwell insert and 500 μL of DMEM supplemented with 10% FBS was placed in the lower chamber. For the invasion assay, a layer of 60 μL Matrigel (354234; Corning, New York, USA) was applied to the bottom of each transwell chamber prior to adding the cell suspension. Next, the Transwell plates were placed in a humidified incubator at 37 °C. After 24 h, inserts were removed from the wells and cells on the upper side of the membrane were gently cleared using a clean wiping tool. Inserts were further treated for 15 min with 4% paraformaldehyde for fixation (P0091; Beyotime, Shanghai, China) and stained with crystal violet for 30 min (C0121; Beyotime, Shanghai, China). The invading and migrating cells on the lower side of the membrane were observed and quantified using a microscope.

### RNA isolation, library preparation, and sequencing

RNA sequencing of HEY A8 cells was conducted. Cells in the experimental group were transfected with ULK2-expressing vector while the control group was transfected with empty vector. RNA was extracted from cultured cells with the aid of TRIzol reagent (15596026; Invitrogen, CA, USA). A Nanodrop 2000 spectrophotometer was employed to assess the quality and concentration of RNA samples. RNA integrity was evaluated using an Agilent 2100 Bioanalyzer and 2100 RNA Nano 6000 assay kit (*Yang et al., 2022*). Poly-A RNA was enriched from eukaryotic total RNA using a TIANSeq mRNA Capture Kit (CB313733565; TIANGEN, Beijing, China). The RNA obtained served as the initial material to generate transcriptome sequencing libraries using the TIANSeq Fast RNA Library Kit. Subsequent to cluster generation, libraries were sequenced on an Illumina platform, which generated 150 bp paired-end reads. Three biological replicates were obtained for all experimental and control groups. The statistical power of this experimental design, calculated in RNASeqPower, was 0.83.

### Third-party data

Third-party data used in this study were obtained from Gene Expression Profiling Interactive Analysis (GEPIA), Kaplan-Meier (KM) plotter, The Cancer Genome Atlas (TCGA) and Human Protein Atlas (HPA) databases. The respective URLs are as follows: http://gepia.cancer-pku.cn (GEPIA database), http://kmplot.com/analysis (KM-plotter database), https://www.cancer.gov/ccg/research/genome-sequencing/tcga (TCGA database) and https://www.proteinatlas.org (HPA database).

### Statistical analysis

Data were processed with the GraphPad Prism tool (Version 8.0, GraphPad Software, San Diego, CA, USA) and expressed as means with standard deviation (SD). For comparison

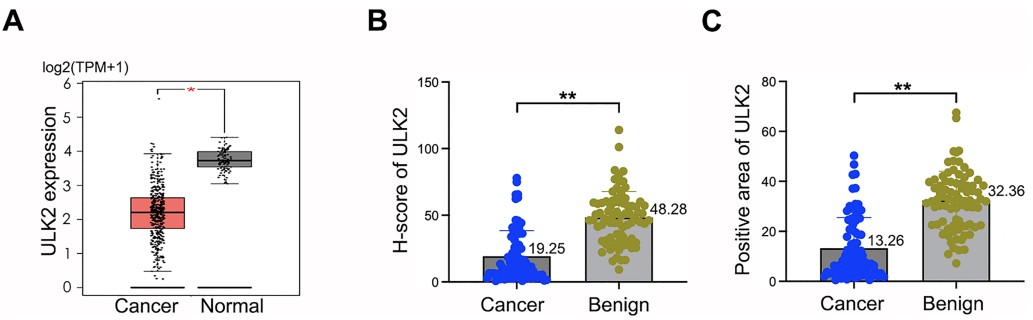

Figure 1 ULK2 was low expressed in epithelial ovarian cancer tissues. (A) The expression level of ULK2 was assessed in ovarian cancer (OC) tissues and normal control (NC) samples obtained from the GEPIA database. The dataset included 426 cases OC tissues from the TCGA database and 88 NC tissues from the GTEx database. TPM: Transcripts per million. *$p < 0.05$. (B) H-score analysis of immuno-histochemistry (IHC) staining for ULK2 obtained from 80 ovarian cancer tissues and 81 benign ovarian tumor samples from our hospital. **$p < 0.01$. (C) IHC results displayed the percentage of ULK2 positive expression in above ovarian cancer tissues and the control samples in our institution. **$p < 0.01$.

between two groups, the t-test was employed and survival rates analyzed using the Kaplan-Meier (KM) assay. *P* values < 0.05 were considered statistically significant.

## RESULTS

### ULK2 was decreased in epithelial ovarian cancer tissues

Analysis of the GEPIA database revealed downregulation of ULK2 in various tumor tissues, including ovarian cancer, relative to the corresponding non-tumor counterparts (Figs. S1A and 1A). The tumor name abbreviations were listed in Table S1. To confirm the conclusions derived from the GEPIA database on ovarian cancer, we collected 80 tissue samples from ovarian cancer patients, along with 81 samples from individuals with benign ovarian cysts or tumors at our hospital. The clinical information of the ovarian cancer patients was presented in Table S2. Thereafter, IHC was conducted to analyze the H-score and the positive rate of ULK2. The findings indicated low expression of ULK2 in ovarian cancer tissues and high expression in benign ovarian cysts or tumors (Figs. S1B and 1B, 1C). Consequently, we deduced that ULK2 expression was significantly diminished in tissues of epithelial ovarian cancer.

### ULK2 was positively related to ovarian cancer patients' survival

Analysis of the GEPIA database revealed that ULK2 exhibited low expression in advanced ovarian cancer and relatively high levels in early-stage cancer (Fig. 2A). A validation assay was further conducted using ovarian cancer samples obtained from our institution. The results revealed low H-Score and positive expression rate of ULK2 in advanced ovarian cancer compared to early-stage cancer (Figs. 2B and 2C). Subsequently, the association between ULK2 and survival outcomes of ovarian cancer patients was examined. Evaluation of the survival period of 557 ovarian cancer patients from the TCGA database indicated a positive link between ULK2 expression and survival (Fig. 2D). Moreover, an analysis of data from 80 epithelial ovarian cancer patients at our hospital yielded data consistent with

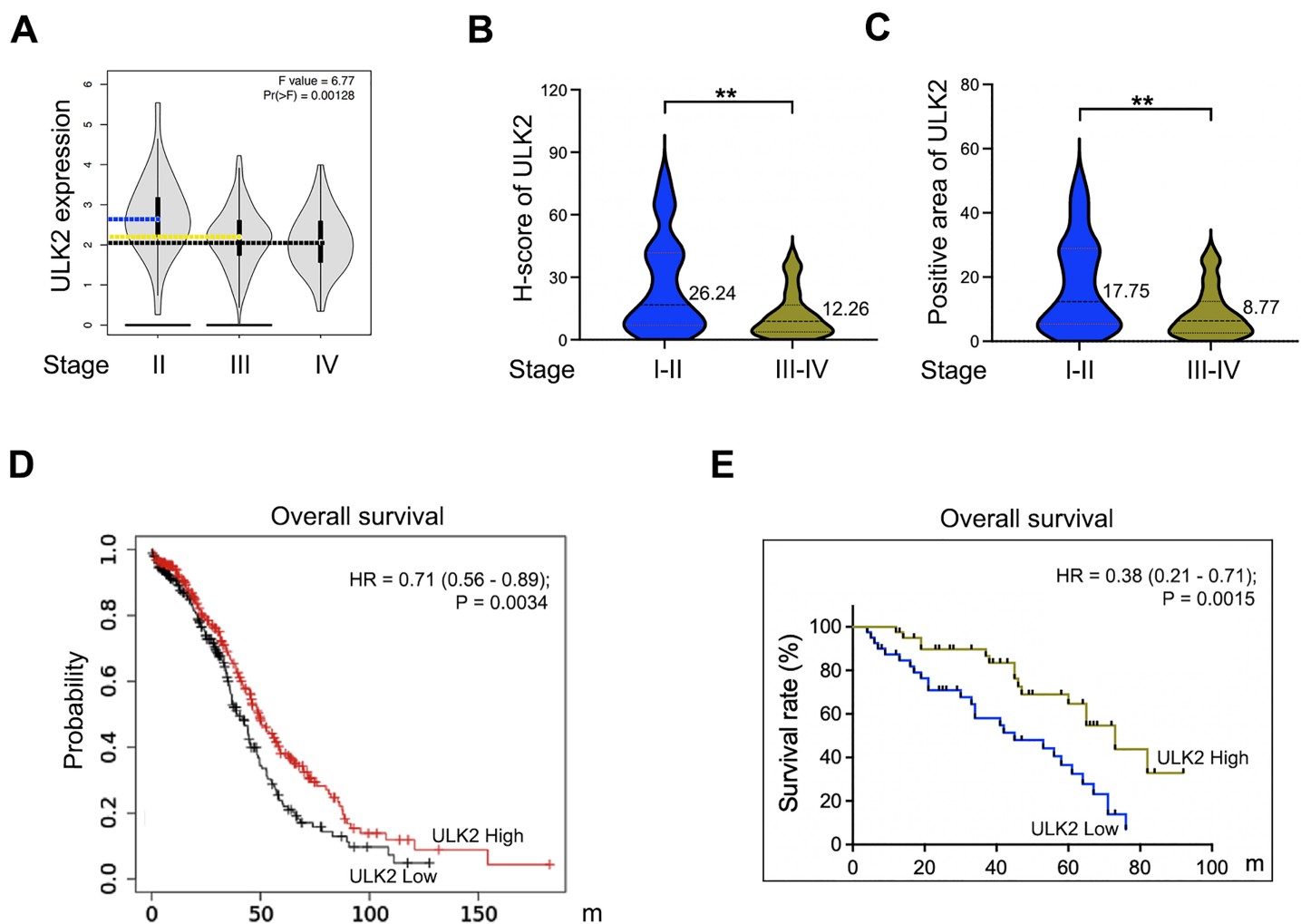

**Figure 2** **The expression of ULK2 in ovarian cancer and its positive correlation with patients' survival.** (A) Analysis of ULK2 expression levels in different stages of ovarian cancer tissues using the GEPIA database. (B) The H-score of ULK2 expression was evaluated in 40 cases each of early and late-stage ovarian cancer at our institution. **$p < 0.01$. (C) The prevalence of ULK2 expression was assessed in 40 cases of early-stage and 40 cases of late-stage ovarian cancer at our institution. **$p < 0.01$. (D) The relationship between ULK2 expression and overall survival was analyzed in 557 ovarian cancer patients from the TCGA database. The survival curve was generated from Kaplan-Meier plotter database. (E) The association of ULK2 and overall survival was examined in 80 ovarian cancer patients at our institution.     

this conclusion (Fig. 2E). Based on the collective findings, we propose that ULK2 functions as a tumor suppressor gene in ovarian cancer and its high expression is associated with more favorable prognosis for patients.

## ULK2 inhibits the proliferation and migration of ovarian cancer cells

Samples of 59 diverse ovarian cancer cell lines from the HPA database were examined for expression levels of ULK2, which revealed low or no expression in numerous cell lines (Fig. 3A). Additionally, Western blot analysis revealed a significant decrease in ULK2 expression in the ovarian cancer cell lines OVCA433 and HEYA8 (Fig. 3B). To ascertain the precise role of ULK2 in ovarian cancer proliferation and migration ability, overexpression of the gene was induced in these two cell lines (Fig. 3C). The CCK-8 assay

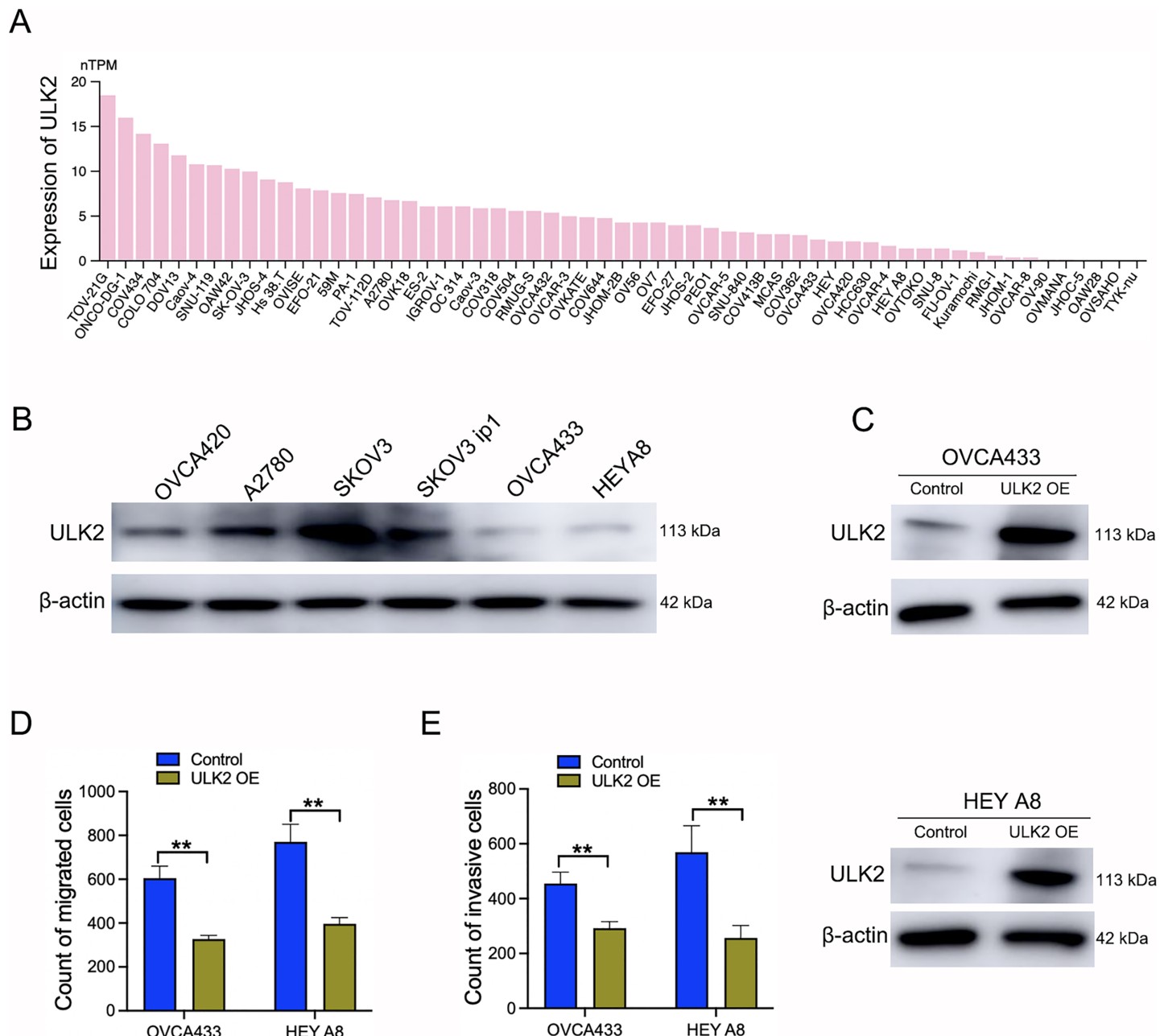

**Figure 3 Effects of ULK2 overexpression on the migration and invasion of ovarian cancer cells.** (A) Expression levels of ULK2 in 59 ovarian cancer cell lines from the Human Protein Atlas (HPA) database. (B) Validation of ULK2 expression levels in ovarian cancer cell lines was performed using Western Blot analysis in the study group. (C) ULK2 overexpression level was determined through Western blot; β-actin was used as the internal control. (D) Statistical results of the transwell migration assay. **$p < 0.01$. (E) Statistical results of the transwell invasion assay. **$p < 0.01$. The Transwell experiments were repeated three times.

indicated that the increased ULK2 levels suppressed the proliferation of ovarian cancer cells in both OVCA433 and HEY A8 cell lines (Figs. S2A and S2B). The transwell assay revealed substantial inhibition of migration and invasion of ovarian cancer cells overexpressing ULK2 (Figs. 3D, 3E and S2C, S2D). Specifically, the migration cell count was inhibited by 45.92% and 50.00% in OVCA433/ULK2-OE and HEY A8/ULK2-OE

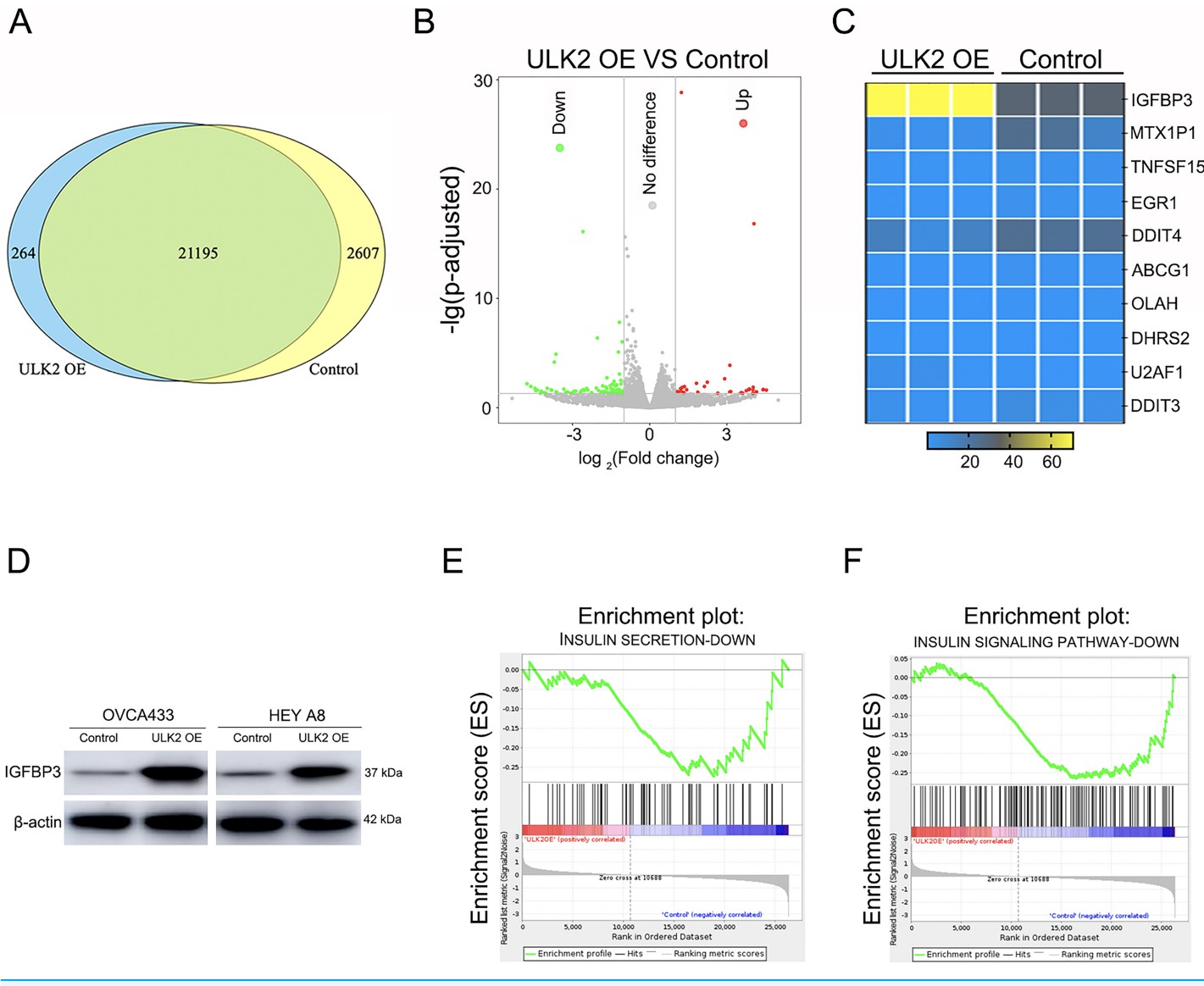

**Figure 4** **The regulation of IGFBP3-mediated insulin pathway by ULK2.** (A) Venn diagram summarizing the results of RNA sequencing; (B) Volcano plot displaying the differential gene expression; (C) Heatmap illustrating the significant differential gene expression identified through RNA sequencing; (D) IGFBP3 expression level was detected by Western blot; β-actin was used as the internal control. (E and F) Gene set enrichment analysis (GSEA) enrichment analysis.

cells, respectively (Fig. 3D). The invasive cell count was significantly inhibited by 35.75% in OVCA433/ULK2-OE cells and by 54.94% in HEY A8/ULK2-OE cells (Fig. 3E). Our results clearly demonstrate that ULK2 markedly suppresses cell invasion and motility in ovarian cancer.

## ULK2 suppresses the insulin signaling pathway by upregulating IGFBP3

To further explore the mechanism underlying the impact of ULK2 on ovarian cancer cell growth and metastasis, RNA sequencing was conducted in HEY A8 cells overexpressing

ULK2 (HEY A8/ULK2 OE) and control cell lines. Overall, 21,195 genes were commonly present in both HEY A8/ULK2 OE and HEY A8/Control cell lines, 264 genes specifically detected in ULK2 OE cells, and 2,607 genes exclusively identified in HEY A8/control cells (Fig. 4A). Within the RNA sequencing data, 119 genes exhibited differential expression, including 35 with increased expression and 84 with decreased expression, in ULK2-overexpressing cells (Fig. 4B, Table S3). Data from Gene Ontology (GO) and Kyoto Encyclopedia of Genes and Genomes (KEGG) pathway analysis are presented in Figs. S3A and S3B. Of particular note, IGFBP3 exhibited the most significant upregulation, as confirmed using statistical analysis (Fig. 4C). Subsequently, Western blot analysis confirmed the upregulating of IGFBP3 induced by ULK2 overexpression (Fig. 4D). The follwing Gene Set Enrichment Analysis (GSEA) of the RNA sequencing data revealed a decrease enrichment in the insulin secretion and insulin signaling pathway under conditions of upregulation of ULK2 (Figs. 4E and 4F). Accordingly, we propose that ULK2 suppresses the migration and growth of ovarian cancer cells through inhibition of the IGFBP3-mediated insulin signaling pathway.

## DISCUSSION

In this study, data obtained from integration of bioinformatics, cellular and molecular biology analyses showed that ULK2 suppresses ovarian cancer cell growth, invasion and migration *via* a mechanism involving IGFBP3 in the insulin signaling axis and thus associated with improved prognosis of ovarian cancer patients.

ULK2, a gene closely linked to autophagy, has a debated role in various tumors. Nevertheless, a growing body of research indicates that ULK2 is downregulated in various tumor types (*Liu & Wei, 2023*), exerting inhibitory effects on tumor initiation and progression (*Shukla et al., 2014*). For example, the ULK2 is reported to suppress cell proliferation and augment sensitivity to cisplatin chemotherapy in non-small cell lung cancer (NSCLC) (*Cheng et al., 2019*). Consistent with this finding, our research demonstrated that ULK2 inhibits the motility and proliferation of ovarian cancer cells and is positively associated with the survival of patients.

At present, limited information is available on the molecular mechanisms of ULK2 other than its traditional role in autophagy. In this study, we innovatively noted an elevation in the IGFBP3 level as well as decreased insulin secretion and insulin signaling subsequent to ULK2 enhancement. Previous studies have identified a minimum of seven insulin-like IGFBPs that have the capability to bind to insulin-like growth factor I (IGF-I) and IGF-II, thereby regulating their activity and signaling functionality (*Izutsu et al., 2022*; *Alterki et al., 2021*; *Hjortebjerg et al., 2021*; *Agerholm et al., 2020*). IGFBPs and IGF play pivotal roles in the IGF signaling pathway, exerting significant impacts on tumor initiation, progression, metastasis, and chemoresistance (*Han & Kim, 2021*; *Lee, Tocheny & Shaw, 2022*; *Lee et al., 2022*; *Du et al., 2017*; *Martinez Baez et al., 2023*) that lead to antitumor effects (*Cai, Dozmorov & Oh, 2020*). For example, *Kuhn et al. (2023)* discovered that IGFBP3 inhibited lung cancer cell invasion and proliferation and was linked to the patients' enhanced survival. Additionally, studies have demonstrated that IGFBP3 regulates additional components of the insulin signaling pathway besides IGF-I/II, such as

protein kinase B (PKB/Akt) and insulin receptor substrate (IRS) proteins (*Liu et al., 2021*; *Wu et al., 2021*; *Li et al., 2021*). Additionally, previous reports have indicated that elevated IGFBP3 levels are correlated with reduced insulin sensitivity and lower insulin secretion rates (*D'Addio et al., 2022*), and inhibition of insulin signaling axis and insulin-like growth factor (*Wang et al., 2023*), in alignment with our findings. Importantly, previous research has demonstrated that IGFBP3 can block angiogenesis by regulating the expression of thrombospondin-1 intracellularly (*Shih, Chen & Torng, 2020*). Moreover, IGFBP3 has been characterized as a suppressor gene of invasion in ovarian cancer and reduced levels of IGFBP-3 are linked to higher tumor grade, advanced stage, and unfavorable clinical prognosis (*Torng et al., 2008*).

Our collective findings suggest that ULK2 can diminish ovarian cancer cell proliferation and migration through inducing an increase in IGFBP3 expression. The pivotal function of IGFBP3 in the insulin signaling axis may account for its regulatory role, representing a novel contributory mechanism to the tumor suppressor effect of ULK2 that is independent of its involvement in autophagy control.

However, our findings also revealed that overexpression of ULK2 influenced multiple signaling pathways linked to tumor progression, including fatty acid biosynthesis, fatty acid elongation and spliceosome, as identified through KEGG pathway analysis (Fig. S3B). These results suggest that ULK2 plays a role in intricate signaling networks in ovarian cancer.

## CONCLUSION

In summary, our current research exploratively discovered that ULK2 suppressed cell migration and invasion of ovarian cancer *via* inhibition of the insulin signaling pathway by promoting upregulation of IGFBP3. The findings suggest that ULK2 expression could therefore serve as a potential indicator for assessing tumor prognosis and as a focus for cancer treatment.

Our research has contributed significant insights into the role of ULK2 in ovarian cancer. However, further studies, particularly *in vivo* experiments, are necessary to fully uncover the impact of ULK2 on the progression of ovarian cancer. These future studies will address this gap in knowledge.

## ACKNOWLEDGEMENTS

We gratefully thank the company of International Science Editing for their assistance in editing our manuscript.

### Funding

The work was supported by the Beijing Great Physician Commonweal Foundation (HX214154 to Weipei Zhu), the National Nature Science Foundation of China (82002804 to Han Xu; 82272641 to Huizhen Sun) and the innovative talent base for master and doctor of hypertensive disorder complicating pregnancy (RCJD2022S06 to Bingyi Yao). The

funders had no role in study design, data collection and analysis, decision to publish, or preparation of the manuscript.

## Grant Disclosures
The following grant information was disclosed by the authors:
Beijing Great Physician Commonweal Foundation: HX214154.
National Nature Science Foundation of China: 82002804, 82272641.
Innovative Talent Base for Master and Doctor of Hypertensive Disorder Complicating Pregnancy: RCJD2022S06.

## Competing Interests
The authors declare that they have no competing interests.

## Author Contributions
- Xiaoxi Chen conceived and designed the experiments, performed the experiments, analyzed the data, prepared figures and/or tables, authored or reviewed drafts of the article, and approved the final draft.
- Changxiang Shao performed the experiments, analyzed the data, prepared figures and/or tables, and approved the final draft.
- Jing Liu performed the experiments, prepared figures and/or tables, and approved the final draft.
- Huizhen Sun performed the experiments, prepared figures and/or tables, and approved the final draft.
- Bingyi Yao analyzed the data, authored or reviewed drafts of the article, and approved the final draft.
- Chengbin Ma conceived and designed the experiments, authored or reviewed drafts of the article, and approved the final draft.
- Han Xu conceived and designed the experiments, analyzed the data, authored or reviewed drafts of the article, and approved the final draft.
- Weipei Zhu conceived and designed the experiments, authored or reviewed drafts of the article, and approved the final draft.

## Human Ethics
The following information was supplied relating to ethical approvals (*i.e.*, approving body and any reference numbers):

Ethical approval was obtained from Changning District Maternal and Child Health Hospital (No. CNFBLLKT-2023-013) prior to the commencement of the project.

## Data Availability
The raw data is available in the Supplemental File.

## Supplemental Information

Supplemental information for this article can be found online at http://dx.doi.org/10.7717/peerj.17628#supplemental-information.

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
