# Peer review of "ULK2 suppresses ovarian cancer cell migration and invasion by elevating IGFBP3"

_PeerJ, doi:10.7717/peerj.17628_

## Round 0.1 · original submission · Major Revisions

Please revise the manuscript as the reviewers suggested.

Reviewer 1 ·

Basic reporting

The authors performed only Gene Set Enriccment Analysis to show that ULK2 reduces insulin secretion, however they did not show real data. If the authors want to conclude that the role of ULK2 in insulin signaling pathway by upregulating IGFBP3 in ovarian cancer cells, they must show real experienced data in vitro or in vivo. Although they concluded that IGFBP3 was high in ovarian cancer, they only show data of a single cell line ULK2. Thus, they must show data using other cell lines.

Even if IGFBP3 affect on insulin expression, the effect of insulin on tumor dissemination is less than that of IGFs. If IGFBP3 was up-regulated in ovarian cancers, the effect of IGFs on migration and invasion might be reduced. The authors must show real data and they discuss these.

In Figure 3A, they showed data of several ovarian cell lines, however the data did not include OVCA433. Why they select OVCA433 ???

In Figures 1A, 1B, and 3A, there is no description of what these vertical axes are and what their credits are. The authors showed same data for multi panels. Thus, Figures 1B, 1D, and 1E are important, however Figures1A and 1C are unnecessary. In addition, Figures 3D and 3F are necessary, however, Figures 3C and 3E are unnecessary.

Experimental design

.

Validity of the findings

.

Additional comments

.

·

Basic reporting

'no comment'

Experimental design

Experimental issues:
1. To compare normal vs tumor ULK2 expression in ovarian cancer, the authors performed a t-test comparison (Figure 1B), finding lower expression of ULK2. However, the proper way to made this comparison is by a paired t-test comparison. It is easy to found in the TCGA data which samples are paired to perform the analysis.
2. The average of the H-scores shown in the Supplementary Table 2 should be accompanied with the standard deviation.
3. About the affirmation that ULK2 is a prognosis marker, the results are not conclusive. The authors demonstrated in the TCGA cohort, and in their own cohort, that ULK2 is significantly associated with the tumor stage, in consequence it is expected that the low expression of ULK2 is associated with prognosis, since I-II stages have better prognosis than III-IV stages. To conclude that ULK2 expression is a good prognostic marker in ovarian cancer it is necessary to complement the KM analysis with multivariate Cox regression analysis.
4. From the transcriptome analysis, the authors concluded that ULK2 modulate negatively the insulin signaling pathway. That conclusion comes from the differential expression of IGFBP3 in ULK2 OE cells. However, their pathway enrichment analysis did not indicate insulin signaling pathway as modulated by ULK2. Also, the authors totally ignored their own results from this transcriptomic analysis, that indicate splicing, fatty acids and other pathways as related with ULK2 overexpression.

Validity of the findings

'no comment'

Additional comments

'no comment'

·

Basic reporting

1. The manuscript is very disorganized and difficult to follow. The sequencing/organization of the data does not follow a logical pattern.
2. This paper has numerous grammar and language issues (Third-party data, etc.), which need to be addressed. The authors get editing help from someone with full professional proficiency in English.

Experimental design

1. Abstract of this paper is poorly written, lacks important information, and conveys a biased picture (there’s no data on metastasis in the text). The conclusion section should be written more comprehensively with key findings, but not simple repetition of the results.
2. The authors should delete the sentences on metastasis and should add more information on IGFBP3 in the Introduction section.
3. In the end of the introduction, the authors should want to leave the reader with a good understanding of why their topic is important, the purpose of the paper, and desire to learn how they conducted the study.
4. Methodology needs to be supplemented, including details on patients’ samples. The authors need to introduce each experiment including the origin of the cells or tissues, the concentration and manufacturer of the reagents respectively and clearly.
5. Protein concentrations might account for some uneven loading on gels. A detergent-compatible reagent needs to be used. Western blot data is difficult to evaluate due to their small size and poor resolution in PDF available review.

Validity of the findings

1. The authors should remove the Lines 216 – 220 from the Results section.
2. Each discussion presented is intriguing, but overall, the work falls short of demonstrating molecular biology at the cellular level on the phenotypes observed. The molecular mechanisms of action between ULK2 and IGFBP3 was not clearly illuminated in this manuscript, thus it requires more experimental research.
3. The figure legends are difficult to read. The over-use of abbreviations is an issue as the readers may not be familiar with them.
4. The legends do not contain enough information to even cursorily understand how the experiments were carried out.

Additional comments

1. The authors should mention limitations that are directly relevant to their research objectives. The paper should end with implications for the interpretations of the results and several recommendations for future research.

---

## Round 0.2 · accepted · Accept

This manuscript has been well revised.

·

Basic reporting

no comment

Experimental design

no comment

Validity of the findings

no comment